# EGFR and p38^MAPK^ Contribute to the Apoptotic Effect of the Recombinant Lectin from Tepary Bean (*Phaseolus acutifolius*) in Colon Cancer Cells

**DOI:** 10.3390/ph16020290

**Published:** 2023-02-14

**Authors:** José Luis Dena-Beltrán, Porfirio Nava-Domínguez, Dulce Palmerín-Carreño, Dania Martínez-Alarcón, Ulisses Moreno-Celis, Magali Valle-Pacheco, José Luis Castro-Guillén, Alejandro Blanco-Labra, Teresa García-Gasca

**Affiliations:** 1Laboratorio de Biología Celular y Molecular, Facultad de Ciencias Naturales, Universidad Autónoma de Querétaro, Av. De las Ciencias s/n. Juriquilla, Querétaro 76230, Querétaro, Mexico; 2Departamento de Fisiología, Biofísica y Neurociencias, Centro de Investigación y Estudios Avanzados del IPN, Av. IPN 2508, Ciudad de México 07360, CdMx, Mexico; 3Departamento de Biotecnología y Bioquímica, Centro de Investigación y Estudios Avanzados del IPN, Km. 9.6 Libramiento Norte, Carretera Irapuato-León, Irapuato 36824, Guanajuato, Mexico; 4Tecnológico Nacional de México, Nstituto Tecnológico Superior de Irapuato (ITESI), Carretera Irapuato-Silao, Km. 12.5, Colonia El Copal, Irapuato 36821, Guanajuato, Mexico

**Keywords:** apoptosis, colon cancer, cytotoxic, EGFR, *Phaseolus acutifolius*, p38, recombinant lectin

## Abstract

Previous works showed that a Tepary bean lectin fraction (TBLF) induced apoptosis on colon cancer cells and inhibited early colonic tumorigenesis. One Tepary bean (TB) lectin was expressed in *Pichia pastoris* (rTBL-1), exhibiting similarities to one native lectin, where its molecular structure and in silico recognition of cancer-type *N-glycoconjugates* were confirmed. This work aimed to determine whether rTBL-1 retained its bioactive properties and if its apoptotic effect was related to EGFR pathways by studying its cytotoxic effect on colon cancer cells. Similar apoptotic effects of rTBL-1 with respect to TBLF were observed for cleaved PARP-1 and caspase 3, and cell cycle G0/G1 arrest and decreased S phase were observed for both treatments. Apoptosis induction on SW-480 cells was confirmed by testing HA2X, p53 phosphorylation, nuclear fragmentation, and apoptotic bodies. rTBL-1 increased EGFR phosphorylation but also its degradation by the lysosomal route. Phospho-p38 increased in a concentration- and time-dependent manner, matching apoptotic markers, and STAT1 showed activation after rTBL-1 treatment. The results show that part of the rTBL-1 mechanism of action is related to p38 MAPK signaling. Future work will focus further on the target molecules of this recombinant lectin against colon cancer.

## 1. Introduction

The use of natural bioactive molecules has opened new perspectives for preventive and therapeutic alternatives against cancer, using either the purification and characterization or recombinant DNA technologies [1,2,3]. Among the widely investigated molecules against cancer are plant lectins, proteins with the ability to bind specifically and reversibly to carbohydrates in the cell membrane, causing an anticancer response [4,5,6,7]. The molecular mechanisms between lectins and transformed cells are mainly related to the interaction with aberrant membrane glycosylation patterns [4] and genetic changes that promote the tumor phenotype. Modifications in glycoprotein’s glycosylation are associated with the mechanism of resistance to therapeutic treatments and its progression [7,8], particularly in colon cancer [9].

Colon cancer is one of the most frequent cancers worldwide [10]. Treatment therapies are based on the use of monoclonal antibodies such as cetuximab and panitumumab, which specifically bind to tyrosine kinase receptors such as the epidermal growth factor receptor (EGRF) and bevacizumab, which neutralize the vascular endothelial growth factor (VEGF) [11,12]. However, more therapeutic molecules are needed for specific and low-cost alternatives. Some plant lectins have been studied for their anticancer effect, and different mechanisms of action based on differential recognition of normal and aberrant glycosylation that play an essential role in cell transformation have been proposed [4,13]. These changes are dependent on the cancer type, and some are associated with colon carcinogenesis, such as the high expression of N-acetylglucosaminyltransferase V (MGAT5) that causes increased β1-6 branching in the extracellular domain of membrane receptors such as tyrosine kinases receptors (TKRs) [9,14].

TKRs are transmembrane glycoproteins constitutively activated in cancer, where the signaling pathways result in the regulation of cell proliferation and apoptosis. The activation of these receptors depends on the binding with their specific ligands that promote dimerization or oligomerization with conformational changes and the recruitment of downstream proteins [15,16]. Evidence shows that using N-glycosylation inhibitors decreases the expression and activation levels of TKRs. These inhibitors provoke the retention of receptors in the endoplasmic reticulum and Golgi apparatus, preventing their participation in carcinogenesis [17].

In particular, the EGFR binding to its ligand, the epidermal growth factor (EGF), has been recognized as essential because it can activate several signaling pathways, including PI3K-AKT (phosphatidylinositol-3-kinase- Protein kinase B or Akt) and MAPK (mitogen-activated protein kinases) as Ras/Raf/MAPK kinase, (MEK)/extracellular-related kinase (ERK), PLCγ/protein kinase C (PKC), p38 MAPKs, c-Jun terminal kinase (JNK), Ca^2+^-calmodulin-dependent protein kinase (CaMK), and the signal transducer and activator of transcription (STAT). These pathways establish internal cross talk that leads to a final response, the modulation of physiological conditions leading to cell proliferation and cell death. However, cancer can be promoted when some signaling pathways are over- or under-active. It is also known that EGFR can be internalized and processed by ubiquitination or its degradation via endocytosis, resulting in temporary downregulation [18,19].

Some legume lectins, such as wheat germ agglutinin (WGA) and concanavalin A (ConA), have been shown to prevent the binding of EGF ligand to its receptor [20]. The recombinant and native lectins from *Artocarpus integrifolia* decrease the levels of EGFR phosphorylation in A431 epidermoid carcinoma cells [21]. The importance of EGFR in the mechanism of action of some lectins has been demonstrated, such as the *Polygonatum odoratum* lectin effect on MCF-7 breast cancer cells, where a decrease in the receptor expression and phosphorylation was time-dependent, and the EGFR gene silencing represented a decrease in apoptosis [22]. Other lectins, such as castor lectin, need to be internalized to cause cell death through ribosomal inactivation. However, various mechanisms have been proposed, including immunomodulation and the antiangiogenic effect [4,23,24].

A Tepary bean (*Phaseolus acutifolius*) lectin fraction (TBLF) was studied and showed a differential cytotoxic effect on cancer cell lines [5]. TBLF exhibited low toxicity, good tolerability, and activation of the immune system in rats [25]. Among the observed adverse side effects were pancreatic hypertrophy and atrophy in intestinal villi and crypts, but with partial recovery in two or six weeks, respectively, after TBLF administration [26,27]. The administration of TBLF decreased the early chemically induced tumorigenesis in the colon in rats, with the activation of apoptotic pathways [6]. The induction of apoptosis and cell cycle arrest in the G0/G1 phase in HT-29 colon cancer cells was observed, involving phosphorylation of p53 (Ser46) and a decrease in BCL-2 expression [28].

These results indicate TBLF’s therapeutic potential against colon cancer; however, TBLF production from Tepary bean seeds often has low yield, low purity, and high cost, making it inconvenient for future therapeutic use [3]. Therefore, a recombinant lectin [29,30] was produced using *Pichia pastoris* yeast [31], and the biochemical characterization, crystallographic structure, and glycan-binding analysis were studied. The recombinant Tepary bean lectin (rTBL-1) showed affinity to β 1-6 branched N-glycans, which are overexpressed in fundamental transmembrane receptors in some types of cancer, such as colon cancer. A high amino acid sequence homology between one native lectin and rTBL-1 was observed, and preliminary results suggested similar cytotoxic effects [31,32].

Here, we studied the apoptotic induction and cell cycle arrest of rTBL-1 on colon cancer cell lines, and the results were compared to TBLF. Additionally, we determined the participation of EGFR and downstream proteins as part of the mechanism of action of rTBL-1.

## 2. Results

### 2.1. rTBL-1 Induces Apoptosis and Cell Cycle Arrest in HT-29 Colon Cancer Cells

In previous studies, it was possible to determine that the LC_50_ for rTBL-1 on HT-29 cells was 0.056 mg protein/mL (0.228 mg lyophilized/mL) [32]. Using this concentration, the apoptotic effect of rTBL-1 after 8 h of incubation was determined and compared against the effect of TBLF (LC_50_ 0.055 mg lectin/mL or 0.402 mg of lyophilized TBLF/mL) previously determined in the same cell line [28]. The results showed that rTBL-1 induces apoptosis in a similar way to TBLF (Figure 1). No significant changes were observed in the early apoptosis with respect to control cells, but there was an increase in late apoptosis (*p* ≤ 0.05) and total apoptosis (*p* ≤ 0.05). Figure 2 shows the Western blot analysis for apoptotic markers where both TBLF and rTBL-1 increased cytochrome C release as part of the mitochondrial apoptotic pathway and cleavage of caspase 3 and PARP-1 as part of the apoptotic cascade. Akt phosphorylation decreased after TBLF and rTBL-1 treatments, which is consistent with the observed apoptotic effect. To rule out a possible necrotic effect, released lactate dehydrogenase activity was determined, and no significant changes were observed in the control cells, which confirmed no necrotic effect. Figure 3 shows the effect of rTBL-1 and TBLF on the cell cycle, where an arrest in the G0/G1 phase and a reduction in the S phase in control cells were observed (*p* ≤ 0.05). Previous results for TBLF also showed arrest in the G1 phase after 8 h of treatment [28].

### 2.2. Apoptosis Induction of rTBL-1 Is Related to EGFR

The apoptotic effect of rTBL-1 on SW-480 cells was determined through apoptosis markers. SW-480 colon cancer cells exhibited high EGFR expression [33] and have shown sensitivity to the apoptotic effect of TBLF [28]. Concentrations ranging from 1.2 to 122 µg protein/mL of rTBL-1 were tested, and it was observed that 61 µg protein/mL was the minimum concentration that showed an apoptotic effect (Figure 4) by cleavage of caspase 3 and poly (ADP-ribose) polymerase-1 (PARP-1) proteins. The cleavage of caspase 3 was confirmed by immunofluorescence staining on SW-480 cells. The opposite was observed in the case of CHO-K1, which does not express EGFR, where the cleavage of caspase 3 necessary for its activation was not observed. The mitogenic effect of rTBL-1 on SW-480 cells was discarded by the determination of the proliferating cell nuclear antigen (PCNA) since some lectins are recognized for their effects as mitogens, such as peanut lectin (PNA) [34].

PARP-1 is related to DNA repair, and its cleavage by caspases is related to the apoptotic process [35]. Histone HA2X phosphorylation was confirmed, as it is related to double-stranded break repair events, indicating the activation of apoptotic processes [36]. On the other hand, an increase in p-53 phosphorylation was observed, where p-53 phosphorylation in different residues (serine 20, 33, 46, 366, and 392 and threonine 81, 304, 377, and 387) is related to the activation of apoptosis [37]. Nuclear fragmentation and apoptotic bodies were observed in SW-480 cells, confirming the apoptosis process. In addition, modification in cell morphology is shown by F-actin detection, where control cells exhibited the characteristic fibers, while treated cells appeared structurally affected. Clusters of F-actin were recognized, possibly referring to actin contraction that occurs within apoptotic bodies and is derived from the activation of proteins such as ROCK during apoptosis [38].

### 2.3. rTBL-1 Induces Modification on EGFR Downstream Proteins on Colon Cancer Cells

The participation of EGFR in the apoptosis induction mechanism of rTBL-1 was studied since rTBL-1 showed an affinity for β1-6-branched N-glycans [31], which are found in the extracellular domain of EGFR [9,39]. EGFR phosphorylation was evaluated in SW-480 cells treated with rTBL-1 in a 24 h concentration–response assay (Figure 5). rTBL-1 induced EGFR phosphorylation in a concentration-dependent manner (high phospho-EGFR/total EGFR ratio), but a decrease in total EGFR was observed (fold change) when using rTBL-1 at 61 and 122 µg protein/mL. In order to understand the kinetics of EGFR phosphorylation and degradation, 61 µg protein/mL of rTBL-1 was tested at 0, 3, 6, 12, 18, and 24 h. EGFR phosphorylation increased significantly at 12 h (*p* ≤ 0.05), determined as phospho-EGFR/total EGFR ratio, but decreased gradually at 24 h (fold change of EGFR).

One of the main EGFR pathways is the phosphatidylinositol-3-kinase (PI3K) and its downstream molecule serine/threonine protein kinase B (PKB or Akt) that inhibits the apoptotic effect and, when overactivated, induces cancer development [40]. Nevertheless, biased or noncanonical pathways of EGFR activation include p38 MAPK proteins, a family of four MAPK involved in proliferation–apoptosis regulation in a context-response manner [41,42].

Figure 6 shows that an increase in phosphorylated Akt (phospho-Akt) was observed only with the highest concentration tested (122 μg/mL of rTBL-1) in a concentration–response curve, but with a significant decrease in both phospho-Akt and total Akt, suggesting that apoptosis could be associated with a decrease in total Akt (fold change). Dephosphorylation and a decrease in total Akt expression have also been observed with other lectins, such as those of the Korean mistletoe [23]. On the other hand, phosphorylation of p38 increased when using 61 and 122 μg/mL of rTBL-1, without affecting total-p-38 (fold change), which coincides with what was observed for EGFR and apoptotic markers.

In order to understand the kinetics of EGFR on the biased downstream proteins, 61 μg/mL of rTBL-1 was tested at 0, 3, 6, 12, 18, and 24 h, and phosphorylation of p38 and STAT1 were determined. The phospho-p38/total p38 ratio increased significantly at 18 h and remained increased until 24 h; however, an increase in phospho-STAT1/total STAT1 ratio was observed at 3 h of rTBL-1 treatment and remained phosphorylated until 24 h. rTBL-1’s effect on STAT1 (Tyr 701) phosphorylation was confirmed after 3 h of treatment, suggesting apoptotic participation of such pathways by rTBL-1 treatment.

Cell membrane decrease in EGFR was observed by immunofluorescence at 6 h of rTBL-1 treatment. Internalization of EGFR has been described as a possible mechanism of degradation [18,43]; therefore, MG132 and chloroquine were used as inhibitors of the proteasomal and lysosomal routes, respectively, in order to determine the mechanism of the receptor degradation by Western blot for total EGFR. The EGFR fold change showed that 75% of the EGFR decreased after rTBL-1 treatment compared with untreated cells; no recuperation was observed when the proteasomal route was inhibited, but a partial recuperation of 50% was observed when the lysosomal route was inhibited. This result suggests that an endosomal internalization was provoked by rTBL-1, which could explain part of the observed apoptotic effect. Direct interaction between rTBL-1 and EGFR was determined by confocal microscopy (Figure 7). Colocalization was observed in yellow; however, lectin not bound to EGFR was observed, possibly by rTBL-1 binding to other membrane receptors.

In order to arrive at an approximation of the interaction between rTBL-1 and EGFR, a docking analysis was performed. A protein–protein docking analysis between rTBL-1 and EGFR was realized on the LZERD protein docking server, and interaction models were analyzed using the software Discovery Studio Visualizer version 21.1.0.20298 (Dassault Systèmes Biovia Corp. Velizy-Villacoublay, France). In this case, the rTBL-1 interaction with EGFR was directly located at the carbohydrate-recognizing domain (CRD) of rTBL-1 with one of four EGFR subunits (Figure 8). The global binding energy of rTBL and EGFR interaction was calculated using the PPCheck server [44].

Representative EGFR-related glycans were chosen, including highly mannosylated- and complex glycans. Figure 9 shows examples of complex EGFR-related 2132 and 2608 N-glycans with a high possibility of interaction with rTBL-1 as 876 (average mass of 4048.6941) that interacted between monomers C and D (https://glyconnect.expasy.org accessed on 27 January 2023) (both with and average mass of 4194.8371). Other N-glycans with a high possibility of interaction were: 2263, 2537, 2874, 3268, and 3414 (Figure 10). Complex N-glycans are specifically recognized by rTBL-1 [31] and, therefore, the interaction zone with small EGFR-related N-glycans given by the docking process was located at the zone between rTBL-1 monomers, indicating a lack of real interaction [45].

## 3. Discussion

Plant lectins have been studied because of their antitumor potential and the ability to identify membrane carbohydrates of cancer cells. The antitumor mechanisms of plant lectins have been demonstrated primarily through apoptosis and autophagy by modifying certain signaling pathways [4]. Our previous studies showed that a Tepary bean lectin fraction (TBLF) exhibited cytotoxic effects on colon cancer cells and inhibited early malignant lesions when colon cancer was induced in rats by azoxymethane. The apoptotic effects were associated with p53 and Akt pathways [6,26,28]. Preclinical studies showed good tolerability of TBLF administration, but some local adverse effects were observed, such as intestinal atrophy and pancreatic hyperplasia with partial recovery in the post-administration period [25,26,27]. TBLF contains at least two bioactive lectins and some other minority proteins [47], and its production involves several purification steps, resulting in a high-cost, low-yield procedure. Therefore, it was necessary to develop a method to obtain a recombinant lectin [29,30,31] rTBL-1 with high homology to one native bioactive lectin [32]. In this sense, we compared the apoptotic effects of both LC_50_ for rTBL-1 versus TBLF on HT-29 colon cancer cells.

Our results showed that rTBL-1 induced similar results to TBLF in the percent of cells in late apoptosis and cell cycle arrest on the G0/G1 phase and the rate of decrease in the S phase in HT-29 cells. Western blot assays showed an increase in caspase 3 and PARP-1 cleavage and the cytochrome C release in HT-29 cells. The results were confirmed for SW-480 cells, where p53 and histone HA2X phosphorylation were also determined. All results pointed to apoptosis induction [36,37]. We also observed rTBL-1-induced morphology changes, such as nuclear fragmentation of F-actin associated with apoptotic changes in treated cells [38,48]. Moreno-Celis reported a decrease in the antiapoptotic protein BCL2 expression on HT-29 after treatment with TBLF and an increase in phosphorylated p53 in serine 46, also related to an apoptotic response. Furthermore, it was observed that TBLF increased caspase 3 activity [28].

Other lectins, such as B*h*L and D*i*L9 lectins, induced cell cycle arrest of G0/G1 on PANC-1 pancreatic cancer cells in a time-dependent manner [49]. *Phaseolus vulgaris* lectin also induced apoptosis and cell cycle arrest in G0-G1 on MCF-7 cells, and mulberry leaf lectin (MLL) had the same effect on MCF-7 and HCR-15 [50,51]. Apoptosis had also been observed when other plant lectins were used on cancer cells; for instance, *Viscum album* var. *coloratum* agglutinin (VCA) induced apoptosis by caspases activation and decreased expression of Akt, NF-κB, and XIAP in colon cancer cells [52] and Korean mistletoe lectins induced apoptosis on HL-60 cells (promyelocytic leukemia cells) [53]. Autophagy has been reported for lectins from *Polygonatum odoratum* on MCF-7 breast cancer cells [22], and its mechanism of action on L929 murine fibrosarcoma cells involves the activation of caspase 8, 9, and 3, and increased levels of expressed FasL and FADD death receptors [54]. Concanavalin A has also shown an autophagy effect on liver cancer cells [55]. ConBr, a lectin extracted from the *Canavalia brasiliensis* seeds, induced autophagy and cell death dependent on caspase 8, and its cytotoxicity on CG glioma cells was associated with MAPKS and Akt pathways modulation [56]. The findings for rTBL-1 regarding the apoptosis and cell cycle indicate that the obtained recombinant lectin retains its cytotoxic bioactivity, which is one of the main characteristics expected to be maintained.

Martinez-Alarcon et al. [31] reported that rTBL-1 recognizes β1-6-branched *N*-glycans present in receptors, such as EGFR, overexpressed in cancer cells; therefore, we evaluated if there was an interaction between rTBL-1 and EGFR as part of the apoptotic mechanism of action. EGFR is a tyrosine kinase receptor used in targeted colon cancer therapy [57]. Canonical signaling EGFR pathway initiates with the ligand binding to the receptor extracellular domain, and its dimerization allows the activation of tyrosine residues in the cytoplasmic domain, triggering several signaling cascades such as Ras/MAPK, PI3K/AKT, and phospholipase C (PLC)/protein kinase (PKC) [58]. However, EGFR activation and its regulation are not fully elucidated because cell stress and other noncanonical pathways can also induce EGFR endocytosis for recycling or degradation processes and p38 MAPK activation [43].

In silico and in vitro studies on cellular targeting of lectins have shown that lectins can interact with cell surface carbohydrates of membrane receptors such as EGFR [22,59]. Lectins’ effect on EGFR has been studied in some human cancer cell lines, such as the epidermoid carcinoma A431 cells. The native lectin of *Artocarpus integrifolia* and its recombinant lectin in *Escherichia coli* decreased EGFR phosphorylation [21]. Another lectin, *Polygonatum odoratum*, in addition to apoptosis, induces EGFR-dependent autophagy, where a decrease in the receptor phosphorylation and eventual degradation of total EGFR were observed in MCF-7 breast cancer cells. A similar pattern occurs with downstream proteins and the activation of apoptotic proteins. siRNAs were used against EGFR, where apoptosis decreased dramatically, confirming that EGFR has a vital role in the induction of apoptosis by *Polygonatum odoratum* lectin [22].

Colon cancer SW-480 cells were used to determine whether the apoptotic effect of rTBL-1 was related to its interaction with EGFR since SW-480 expresses a higher level of EGFR than HT-29 cells [33]. Western blot and immunofluorescence analyses using different concentrations of rTBL-1 on SW-480 cells for 24 h showed that 61 µg protein/mL was the minimum concentration with apoptotic effect, while no effect was observed in CHO-K1 cells (EGFR-). On the basis of the canonical signaling EGFR pathway and the previous reports for TBLF [6,28,31], we evaluated rTBL-1 effects on EGFR activation and downstream proteins using a concentration–response assay (1.2 to 122 μg/mL) at 24 h as well as on noncanonical signaling as p38 and STAT1 activation.

rTBL-1 (61 and 122 μg/mL) significantly increased phospho-EGFR with a significant decrease in total EGFR (high phospho-EGFR/total EGFR ratio), showing that most of the EGFR protein was involved in the process and also that EGFR decreased, possibly as a result of a degradation process. It has been shown that *Polygonatum odoratum* and jacalin lectins also affect EGFR decrease [21,22]. It is known that, as a consequence of saturation signaling, EGFR can be recycled by endocytosis [43]. To better understand the mechanism of EGFR degradation, two inhibitors were used, MG132 (10 nM), which inhibits the proteasomal degradation pathway, and chloroquine (250 µM), for inhibiting the lysosomal pathway. According to the results, EGFR degradation begins after 18 h of treatment; the inhibitors were administered 6 h prior to completing 24 h of rTBL-1 treatment (61 μg/mL) to avoid excessive EGFR degradation. The results indicated that EGFR degradation was at 75% compared to untreated cells, and it was possible to recover the EGFR level up to 50% after the lysosomal route, but no recovery was observed after inhibiting the proteasomal way. This result suggests that the partial recovery of EGFR was carried out mainly by the lysosomal route; however, the result needs further investigation. It is considered that the proteasome degradation pathway is related to receptor recycling and is a temporary effect, as shown with radiotherapy exposure [60]. Nevertheless, the endosomal route represents the primary mechanism to maintain the balance of EGFR signaling and undergo proliferative effects or apoptosis pathways [61]. The phosphorylation kinetics for EGFR using rTBL-1 (61 μg/mL) was tested from 0 to 24 h. Activation of EGFR started at 3 h, and the maximum level was observed at 12 h, after which a partial decrease was observed, possibly due to receptor degradation.

One canonical pathway of EGFR is through Akt signaling; however, no effect on phospho-Akt was observed using rTBL-1 (61 µg protein/mL). Using the highest concertation (122 µg protein/mL), a significant decrease in both phospho-Akt and total Akt was observed, although the phospho-Akt/total Akt ratio was high. The results point to Akt degradation; therefore, Akt signaling might not be involved in the proapoptotic effect since Akt phosphorylation is related to cell proliferation induction and apoptosis inhibition [40]. The apoptotic effect of EGFR can also lead to biased signaling downstream by STAT1 and STAT3 activation in cancer cell lines [19,62]. The transcription factor, STAT1 in particular, is activated by p38 MAPK [41,63]; therefore, the effect of rTBL-1 was assayed on p38 and STAT1 phosphorylation. After the dose–response assay for 24 h, an increase in phospho-p38 was determined without a decrease in total p-38 but with a high phospho-p38/total p38 ratio, suggesting that not all p38 isoforms are involved in the apoptotic process [64]. The kinetics assay showed an increase in phospho-p38 at 18 h, partially maintained until 24 h.

The p38 belongs to the MAPK family and has been associated with the induction of apoptosis since it can induce nuclear translocation of p53 and function as a transcription factor for apoptotic proteins such as Bak, Noxa, Apaf, and Puma, in response to oxidative stress [65]. The phosphorylation p38 was also observed when human carcinoma cells were treated with jacalin lectin [21] and a recombinant lectin from *Artocarpus nitidus subsp. Lingnanensis* (ALL) induced apoptosis by p38 activation in human B-lymphoma cells [66]. *Momordica charantia* lectins induced apoptosis in nasopharyngeal cancer cell lines, promoting phosphorylation of p38 and other kinases, such as JNK [67]. *Polygonatum cyrtonema* lectin promotes cytotoxicity by apoptosis in human melanoma cells (A375) as a result of a mitochondrial imbalance that leads to increased oxidative stress and activation of the p38 MAPK-p53 pathway [68]. *Morus alba* leaf lectins were shown to have proapoptotic effects in MCF-7 breast cancer cells by increasing p38 phosphorylation [69]. It has been widely observed that p38 is activated in response to various environmental and cellular stress signals and is currently considered a therapeutic target for various types of cancer [70].

STAT1 is one of the seven members of the signal transducers and transcription activators (STAT-1, STAT-2, STAT-3, STAT-4, STAT-5α, STAT-5β, and STAT-6). STAT1 has been shown to be an apoptotic inducer activated by caspases, as its phosphorylation in tyrosine-701 or serine-727 is related to the apoptotic effects. It has also been recognized as a modulator of p53 [71,72,73]. The kinetics showed an increase in the phospho-STAT1/total STAT1 ratio from 3 h to 24 h, probably because STAT1 can be activated by more than one pathway, and the final response depends on the cellular context [74,75]. The same concentration of rTBL-1 tested (61 µg protein/mL) showed an increase in phospho-STAT1 (Tyr 701) after 3 h.

Using immunofluorescence analysis, it was possible to observe the colocalization of rTBL-1 and EGFR. The rTBL-1 localization in the membrane surface without EGFR signal was also noted, suggesting that other surface molecules may be involved in the cellular interactions of rTBL-1. EGFR is an important glycoprotein for cancer biology with a significant number of glycosylation sites (7–13 N-glycosylations in the extracellular domain); it is highly expressed in different cancers and is considered a proteomic marker [76,77,78,79]. In addition, EGFR-specific N-glycosylations are correlated with the disease state/progression of colorectal cancer [80], thus, EGFR-related N-glycans were considered for the docking process. After the docking analysis, it was determined that rTBL-1 could recognize EGFR N-glycans by hydrogen bonds and hydrophobic interactions. Interactions between rTBL and EGFR could occur at different sites, depending on the glycosylated state of EGFR and other factors, even with protein–protein interactions. Nonetheless, the interaction was located directly at the rTBL-1 CRD.

Our results show that rTBL-1 affects human colon cancer cells by affecting EGFR pathways, mostly by partially degrading the receptor and activating p38 MAPK signaling. Figure 11 shows a proposed mechanism of rTBL-1 action as a result of its interaction with EGFR.

## 4. Materials and Methods

### 4.1. Tepary Bean Lectin Fraction (TBLF)

TBLF was obtained from Tepary bean seeds. Briefly, the seeds were finely ground and the flour was degreased with CHCl_3_/MeOH (3:1) washes until the filtrate was clear. Total proteins were extracted using Tris-HCl pH 8.0 at 4 °C, and gradual precipitation with ammonium sulfate from 40% to 60% (*w*/*v*) saturation was followed by dialysis. Finally, the protein extract was separated by molecular weight exclusion chromatography using Sephadex G-75 columns [5].

### 4.2. rTBL-1 Production

Pichia pastoris SMD1168H yeast (provided by Dr. Elaine Fitches, Department of Biosciences, Durham University, Durham DH1 3LE, UK) was used for the heterologous expression and purification of rTBL-1 [31,32]. The secreted protein in the culture medium was centrifuged (30 min, 7500× *g*, 4 °C), and the supernatant was clarified and subsequently filtered with 2.7 and 0.7 μM glass fiber filters (Whatmann, Maidstone, UK). rTBL-1 was purified from the supernatant using HisTrap HP nickel affinity columns (GE Healthcare, Maidstone, UK) and, finally, dialyzed and lyophilized. The protein was quantified by the Bradford method [81], and SDS-PAGE was performed [82].

### 4.3. Cell Culture

HT-29 (ATCC HTB-38) and SW-480 (ATCC CCL-228) human colon cancer cells were used. To study the participation of EGFR, and in the absence of EGFR cells, the CHO-K1 cell lines (ATCC CCL-61) were used [83]. Cells were seeded in 6-well plates with Dulbecco’s modified Eagle’s medium (DMEM, Gibco, New York, NY, USA), supplemented with 10% Fetal Bovine Serum (FBS, PAN Biotech # P30-3306, Aidenbach, Germany) and 1% antibiotics (#15240062, Gibco, New York, NY, USA). Cultures were incubated in a humidified atmosphere with 5% CO_2_ at 37 °C, with media changing every two days until the desired confluence.

#### 4.3.1. rTBL-1’s Effect on Apoptosis and Cell Cycle

rTBL-1 LC_50_ [32] was tested to determine its effect on apoptosis and cell cycle. After 8 h of treatment, 1 × 10^6^ cells were taken and centrifuged at 300× *g* for 5 min, washed with PBS 1x, and fixed with 70% ethanol for 3 h at −20 °C; after which, 200 µL of the fixed cell suspension was taken, centrifuged at 300× *g* for 5 min, and 3 washes were carried out with PBS 1×. The cell cycle was determined using the Muse Cell Cycle Kit (Cat. No. MCH100106, Darmstadt, Germany) following the manufacturer’s instructions. Determination was performed using the Muse Cell Analyzer Cytometer Cytometer (Merk Millipore, Darmstadt, Germany). To determine the apoptotic effect of rTBL-1, the previously described procedure was carried out until the cell suspension was obtained. The Muse Annexin V & Dead cell kit was used (Cat. No. MCH100105, Darmstadt, Germany) following the manufacturer’s instructions, and determination was carried out using the Muse Cell Analyzer Cytometer. Results were compared against TBLF LC_50_ in at least three independent experiments.

#### 4.3.2. Necrosis Assay by Lactate Dehydrogenase Determination

Lactate dehydrogenase activity in the culture medium was determined to evaluate the effect on necrosis. Cells were seeded (1 × 10^4^ cells/well) in a 96-well plate in triplicate for each treatment: blank was included incubating only with 0.5% BSA-DMEM without cells for 8 h at 37 °C, negative control (NC) consisted of 200 µL of 0.5% BSA-DMEM, positive control (PC) consisted of seeding 1 × 10^4^ cells in 200 µL of 1% Triton X-100 DMEM, and the treated groups (TGs) with rTBL-1 LC_50_ and TBLF LC_50_ in 0.5% BSA-DMEM. After 8 h of incubation, samples were assayed following the commercial kit instructions (LDH-Cytotoxicity Colorimetric Assay, BioVision # K311-400, Milpitas, CA, USA). Cytotoxicity was calculated at 495 nm in three independent experiments; each triplicate was adjusted to the blank, and the following formula was used:Cytotoxicity (%) = (OD of TG − OD of NC)/(OD of PC − OD of NC) × 100

### 4.4. rTBL-1’s Effect on EGFR and Downstream Proteins

Once the cytotoxic effect of rTBL-1 was confirmed, the effect on EGFR was evaluated in SW-480 colon cancer cells, since they express a greater amount of the receptor than HT-29 cells [33]. rTBL-1 recognizes N-glycosylations with β 1–6 branches, and this type of glycosylation is present in EGFR [31]. The cells (3 × 10^4^) were cultured in 6-well plates in DMEM/F12 medium (GIBCO, Grand Island, USA) for 24 h, and rTBL-1 concentrations were tested: 1.2, 12, 61, and 122 μg/mL of protein. Cells were treated with RIPA lysis buffer (150 mM NaCl, 1% NP-40, 0.5% deoxycholic acid, 0.1% SDS, 50 mM Tris, pH 8.0) supplemented with protease inhibitors (Sigma Aldrich, #P9340, Burlington, MA, USA). Cell lysates were sonicated (Sonics Vibra Cell, Newtown, CT, USA), centrifuged at 10,000 rpm for 10 min at 4 °C, and the supernatant was recovered. Protein concentration was calculated with bicinchoninic acid assay (Pierce BCA Protein Assay Kit # 23225, Thermo Fisher Scientific, Waltham, MA, USA) using BSA as standard. SDS-PAGE was performed using 30 µg of protein with 4X sample buffer (LDS Sample Buffer #1772836, Novex by Life Technologies, Carlsbad, CA, USA) in 1.5 mm glasses with 10% polyacrylamide gels. Transference to the nitrocellulose membrane (Cat #1620115, Bio-Rad Laboratories, Inc., Hercules, CA, USA) was carried out for 2.5 h at 0.45 amperes at 4 °C. Ponceau red was used to evaluate protein transference, and the membrane was blocked with Blotting-Grade Blocker (Cat. No. 1706404, Bio-Rad Laboratories, Inc., Hercules, CA, USA) for 1 h at room temperature under constant agitation.

Western blot was performed by incubating overnight at 4 °C with the following primary antibodies: cleaved PARP1 (Cell Signaling Technology #9541; Danvers, MA, USA); PARP-1 (F-2) (sc-8007, Santa Cruz Biotechology, Dallas, TX, USA); cleaved Caspase 3 (Cell Signaling Technology #9661;Danvers, MA, USA); phospho-p38 MAPK (Cell Signaling Technology #4511; Danvers, MA, USA); total p38 MAPK (Cell Signaling Technology #8690; Danver, MA, USA) phospho-Akt (Cell Signaling Technology #9271; Danvers, MA, USA); total Akt (Cell Signaling Technology #9272; Danvers, MA, USA); PCNA (Cell Signaling Technology #13110; Danvers, MA, USA); phospho- EGFR (Cell Signaling Technology #2234; Danvers, MA, USA); total EGFR (Cell Signaling Technology #4267; Danvers, MA, USA); Santa Cruz Biotechology, Dallas, TX, USA); phospho-H2A (Abcam #ab81299; Cambridge, UK); phospho-p53 (Ser15) (Cell Signaling Technology #9284; Danvers, MA, USA); cytochrome c (7H8) (sc-13560; Santa Cruz Biotechology, Dallas, TX, USA); Anti-His Tag Pierce MA121315; Thermo Fisher Scientific, Waltham, MA, USA); phospho-STAT1 (Cell Signaling Technology #7649; Danvers, MA, USA), total STAT1 p84/p91 ((E-23): sc-346, Santa Cruz Biotechology, Dallas, TX, USA); phospho-H2A, Abcam # ab81299; Cambridge, UK); GAPDH (sc-32233; Santa Cruz Biotechology, Dallas, TX, USA). Finally, the membranes were revelated by ECL (SuperSignal West PicoPlus, Thermo Fisher Scientific, Waltham, MA, USA). All assays were made in at least three independent experiments. Western blot images were analyzed with ImageJ software [84].

EGFR degradation analysis was carried out using two inhibitors, MG132 (Sigma-Aldrich #C2211, St. Louis, MO, USA) for proteasome route inhibition [85], and chloroquine diphosphate (Sigma Life Science #C6628-256, St. Louis, MO, USA) for the lysosomal route inhibition [86]. In order to avoid excessive EGFR degradation, cells were treated with rTBL-1 (61 μg/mL) for 18 h, and, after which, MG132 10 nM or chloroquine 250 µM was added to cell cultures for an additional 6 h. Total EGFR was determined by Western blot.

### 4.5. Immunofluorescence Analyses

SW-480 cells (1 × 10^4^) were seeded on glass coverslips, grown to 70% confluence, and treated with rTBL-1 (61 µg/mL) at 37 °C for different lengths of time until 24 h had passed. After which, cells were fixed for 10 min with 4% paraformaldehyde in 1xPBS and permeabilized with 0.5% Triton-X100 in 1xPBS for 10 min at room temperature. Cells were blocked with 1% BSA in PBS for 1 h at room temperature. Subsequently, the cells were incubated overnight with the tested antibody: anti-EGFR (Cell Signaling Technology, #4267), anti-his (Anti-His Tag: Pierce MA121315; Thermo Fisher Scientific, Waltham, MA, USA), cleaved caspase 3 (Cell Signaling Technology #9661; Danvers, MA, USA), and α-tubulin (DM1A) (sc-32293, Santa Cruz Biotechnology, Dallas, TX, USA), followed by a 1 h incubation at room temperature with anti-rabbit conjugated to Alexa 647 (A21244, ThermoFisher Scientific Waltham, MA, USA) (1:1000 dilution) or anti-mouse conjugated to Alexa 488 (A-11001, Thermo Fisher Scientific, Waltham, MA, USA) (1:1000 dilution). Finally, cells were mounted with Fluoroshield/DAPI media (Sigma Aldrich, #F-6057, Burlington, MA, USA), and images were captured using an LSM 510+ Laser Multi Photonic confocal microscope (Carl Zeiss, Baden-Württemberg, Germany) using 40× immersion objective. Phalloidin-IFluor 555 reagent was used to detect F-actin, after which the cells were blocked (ab176756, Abcam, Cambridge, UK) and images were captured at 100× magnification.

### 4.6. Docking Analysis of rTBL and EGFR

A docking analysis for the interaction between rTBL-1 and EGFR was performed with the Local 3D Zernike descriptor-based protein Docking (LZERD) web server (https://lzerd.kiharalab.org accessed on 27 January 2023). GlyConnect platform (https://glyconnect.expasy.org/ accessed on 27 January 2023) from the Expasy portal of the SIB Swiss Institute of Bioinformatics [87] was used to identify EGFR-related glycans where 26 of 52 EGFR-related glycans were chosen. The structure of N-glycan models was created using a “Build Via Text” tool from the GLYCAM web server (https://glycam.org accessed on 27 January 2023), based on the monosaccharide sequence on the GlyConnect platform. All conformers were considered for the docking process.

Docking of EGFR-related glycans with rTBL-1 was performed according to LZERD platform requirements [88]. A list of 500 ranked models with their respective structure files was obtained for each docking job, and the models with higher Ranksum values were chosen. Otherwise, docking models whose interaction was located at the intermeric zone of the quaternary structure of rTBL-1 were discarded because of steric hindrance between monomers and because it was recently correlated to a lack of real interaction in the case of a model of the non-fetuin Tepary bean [45]. Models that fulfilled quality criteria were analyzed using the Protein–Ligand Interaction Profiler server (PLIP server—https://plip-tool.biotec.tu-dresden.de/plip-web/plip/index accessed on 27 January 2023) [89].

### 4.7. Statistical Analysis

Statistical analyses were carried out using T-tests (*p* ≤ 0.05) to compare rTBL-1 vs. TBLF and ANOVA post hoc (Dunnett, *p* ≤ 0.05) to compare the treated vs. control groups. The images were analyzed using the ImageJ ^®^ software [83], and graphics were performed using the Prism-GraphPad software v.9.5.0 (La Jolla, CA, USA).

## 5. Conclusions

The recombinant lectin rTBL-1 obtained from *Phaseolus acutifolius* in *Pichia pastoris* yeast can induce apoptosis and cell cycle arrest similarly to the native lectin fraction TBLF in human colon cancer cells, which indicates that it retains the desired biological properties. The advantage of using rTBL-1 lies in the fact that TBLF is a semi-pure fraction of Tepary bean lectins, unlike rTBL-1, which is a single lectin. In addition, the process for obtaining TBLF is more expensive, low yields are obtained and takes longer. In contrast to rTBL-1, all these factors make the use of TBLF as a potential therapeutic agent not feasible, thus, the development of rTBL-1 with a therapeutic approach is necessary. By studying the mechanism of action and the effect on the EGFR canonical and biased signaling pathways, it was found that the proliferative signaling of Akt activation is inhibited. EGFR also decreased after rTBL-1 treatment and was partially degraded by the lysosomal route. Phospho-p38 MAPK increased simultaneously with apoptotic markers, which makes p38 MAPK a promising mechanism of action, especially since STAT1 is also activated after treatment with rTBL-1. In silico analysis showed that rTBL-1 can interact with different EGFR N-glycans, however, and this interaction partially explains the apoptotic effect of rTBL-1, and, although it was colocalized with EGFR in the cell membrane, part of the recombinant lectin was found in different membrane surface locations, hitherto undescribed. Additionally, the data show a decreased of total EGFR and activations of p38 in a concentration- and time-dependent manner. As p38 has been recognized as an antitumorigenic factor, it will be important to continue exploring other routes of activation and downstream protein participation in order to find target molecules for the anticancer effect of rTBL-1.

## Figures and Tables

**Figure 1 pharmaceuticals-16-00290-f001:**
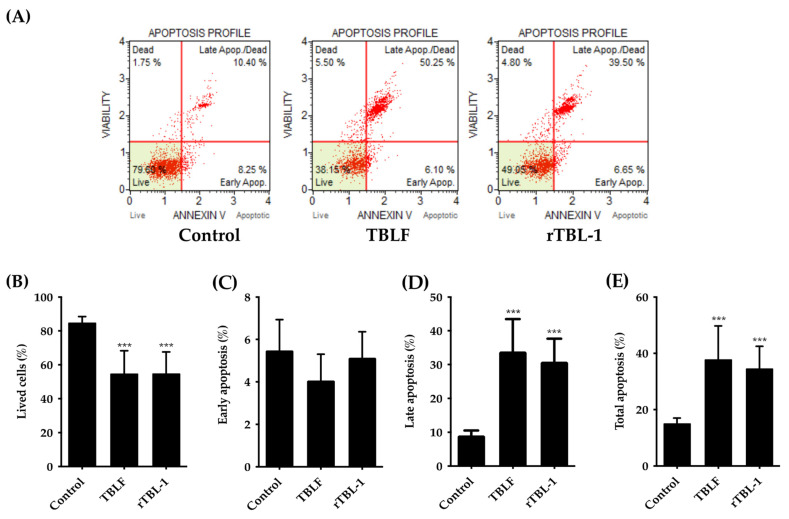
rTBL-1 induced apoptosis in HT-29 cells. Cells were treated with TBLF-LC_50_ or rTBL-1-LC_50_ for 8 h. (**A**) Flow cytometry representative dot plots for apoptosis. (**B**) Cells viability percentage. (**C**) Early apoptosis percentage. (**D**) Late apoptosis percentage. (**E**) Total apoptosis percentage. (***) indicate statistically significant differences between concentrations of both lectins respect to control cells (Dunnett *p* ≤ 0.001). Results of at least three independent experiments are shown.

**Figure 2 pharmaceuticals-16-00290-f002:**
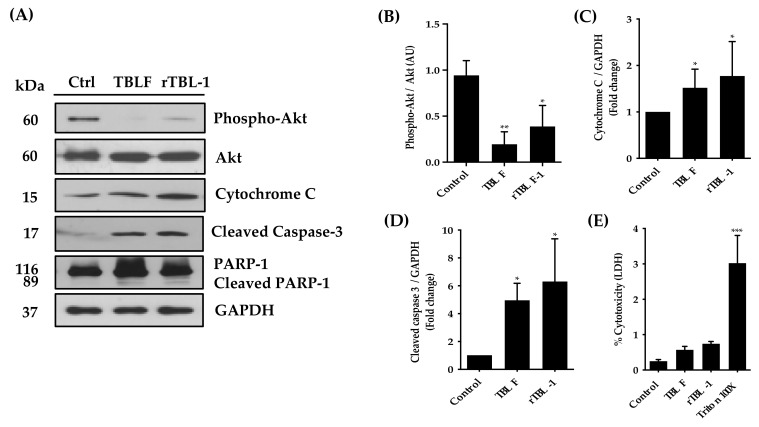
The apoptotic effect of TBLF and rTBL-1 was confirmed by Western blot analysis and LDH release. (**A**–**D**) Western blot and densitometrical analysis for proliferation or apoptosis markers. (**E**) Necrosis determination as a percentage of lactate dehydrogenase release. Dunnett post hoc analyses (*) *p* ≤ 0.05, (**) *p* ≤ 0.01, and (***) (*p* ≤ 0.001) indicate significant differences with respect to control cells. Results of at least three independent experiments are shown.

**Figure 3 pharmaceuticals-16-00290-f003:**
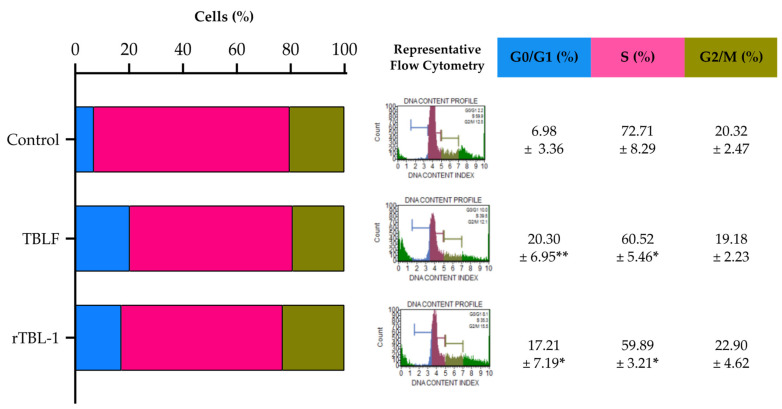
rTBL-1 induced G0/G1 and S cycle cell arrest on HT-29 cells. Cells were treated with TBLF-LC_50_ or rTBL-1-LC_50_ for 8 h. Graphic representation for cell cycle changes, representative flow cytometry assays, and results of at least three independent experiments shown in percentage. One-way ANOVA was performed for each cell cycle phase. (*) *p* ≤ 0.05 and (**) *p* ≤ 0.01 indicates a significant difference (Dunnett) with respect to control cells.

**Figure 4 pharmaceuticals-16-00290-f004:**
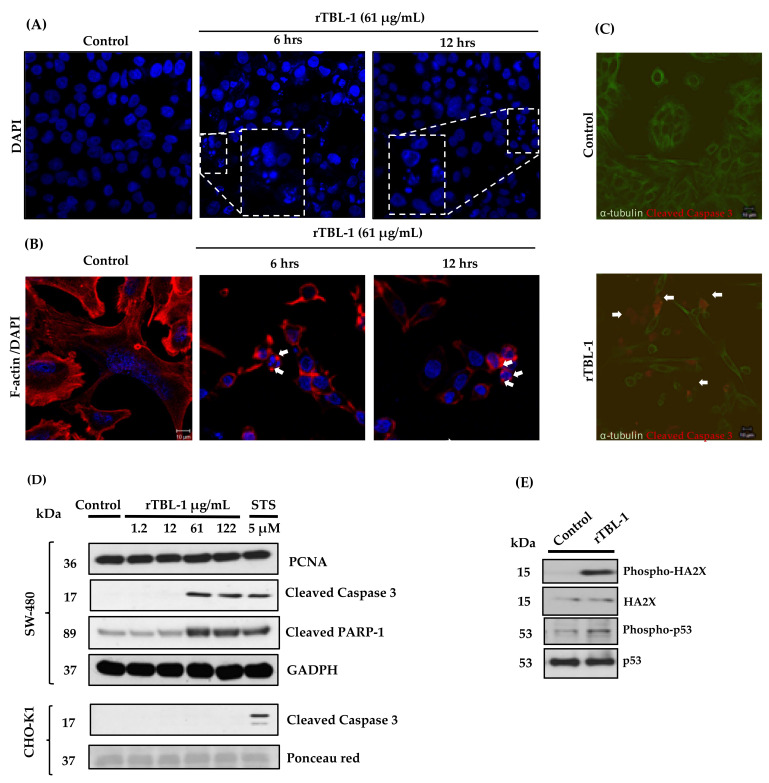
Apoptosis induction of rTBL-1 on SW-480 colon cancer cells. (**A**) DAPI-stained cells showed nuclear fragmentation of SW-480 cells treated with 61 mg/mL rTBL-1 at 6 and 12 h by confocal microscopy at 40× magnification. (**B**) Apoptotic bodies (white arrows) and F-actin changes were detected with phalloidin of SW-480 cells treated with 61 mg/mL rTBL-1 at 6 and 12 h by confocal microscopy at 100× magnification. (**C**) Immunofluorescence for cleaved caspase 3 shown by white arrows (Alexa 594) and α-tubulin (Alexa 488) on SW-480 cells treated with 61 µg protein/mL during 24 h by confocal microscopy at 40× magnification. (**D**) Western blot analysis of apoptotic markers of treated SW-480 (EGFR^+^) and CHO-K1 (EGFR^-^) cells at different concentrations of rTBL-1. (**E**) Western blot of HA2X and p53 phosphorylation on SW-480 by 61 µg/mL rTBL-1.

**Figure 5 pharmaceuticals-16-00290-f005:**
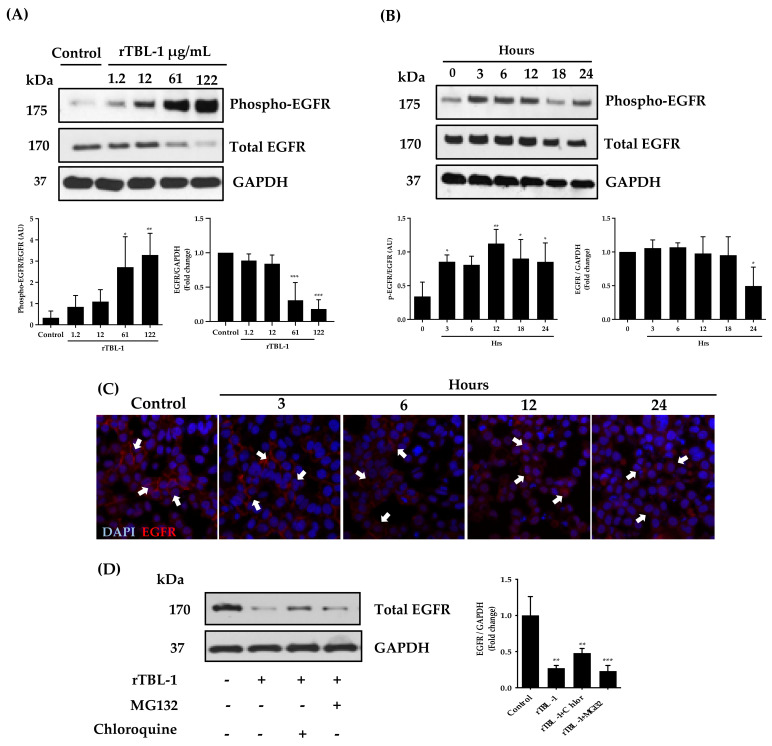
Effect of rTBL-1 on EGFR in SW-480 colon cancer cells. (**A**) Representative Western blot of phosphorylated and nonphosphorylated EGFR proteins after treatment with 1.2, 12, 61, and 122 μg protein/mL of rTBL-1 for 24 h. The phosphorylation level of each protein was determined by the ratio of the density of the phosphorylated EGFR band over the total EGFR band. (**B**) Representative Western blot of phosphorylated and total EGFR when cells were treated with 61 μg protein/mL of rTBL-1 for 0, 3, 6, 12, 18, and 24 h. (**C**) Immunofluorescence analysis for EGFR on SW-480 cells treated for 3, 6, 12, and 24 h with rTBL-1 61 μg protein/mL. (**D**) Analysis of EGFR degradation by proteasomal route (MG132 10 nM) or lysosomal route (chloroquine 250 μΜ). Cells were treated with 61 μg protein/mL of rTBL-1 for 18 h, and, after which, inhibitors were added for an additional 6 h. Dunnett post hoc analyses (*) *p* ≤ 0.05, (**) *p* ≤ 0.01, and (***) *p* ≤ 0.001 indicate a significant difference compared with control cells.

**Figure 6 pharmaceuticals-16-00290-f006:**
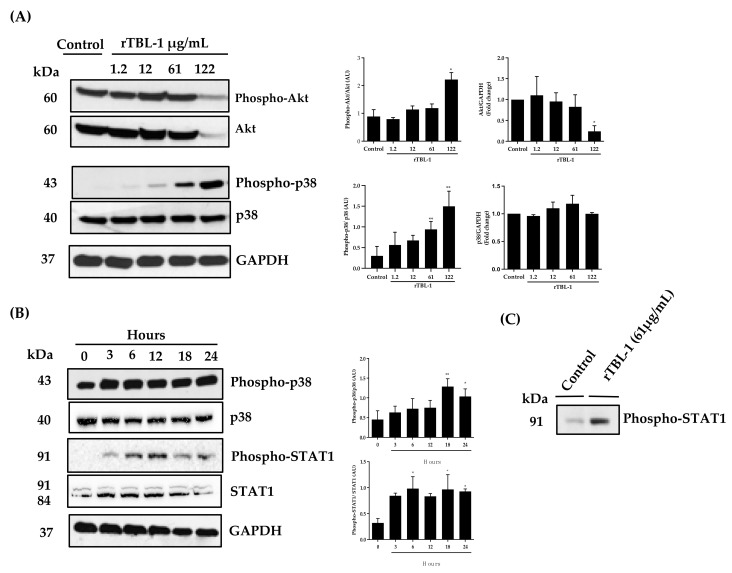
Effect of rTBL-1 on p38 and STAT1 activation in SW-480 colon cancer cells. (**A**) Representative Western blot of phosphorylated and nonphosphorylated Akt and p38 proteins after treatment with 1.2, 12, 61, and 122 μg protein/mL of rTBL-1 for 24 h. The phosphorylation level of each protein was determined by the ratio of the density of the phosphorylated band over the total band. (**B**) Representative Western blot of phosphorylated and total p38 and STAT1 proteins when cells were treated with 61 μg protein/mL of rTBL-1 for 0, 3, 6, 12, 18, and 24 h. (**C**) Effect of rTBL-1 on STAT1 (Tyr 701) phosphorylation after 3 h in 61 μg protein/mL of rTBL-1. Dunnett post hoc analyses; (*) *p* ≤ 0.05 and (**) *p* ≤ 0.01 indicate the significant difference compared with control cells.

**Figure 7 pharmaceuticals-16-00290-f007:**
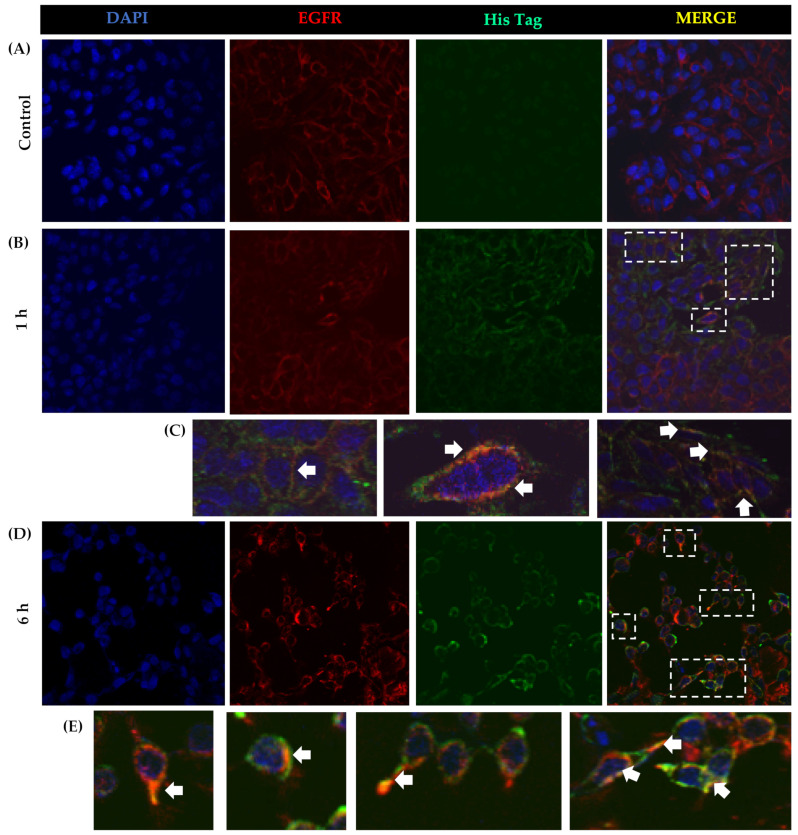
Colocalization of rTBL-1 and EGFR in SW-480 cell membrane. SW-480 cells were treated with rTBL-1 (61 μg/mL) at 0, 1, and 6 h, followed by analysis by immunofluorescence and imaging by confocal microscopy at 40× magnification. (**A**,**B**,**D**) Nuclei DAPI staining; Anti-EGFR and Anti-His were detected through Alexa 647 and Alexa 488, respectively. (**C**,**E**) Amplified sections of merge images (white dotted lines) and white arrows show colocalization rTBL-1-EGFR. Green color shows membrane-bound rTBL-1 not colocalized with EGFR. Representative images from at least two independent experiments.

**Figure 8 pharmaceuticals-16-00290-f008:**
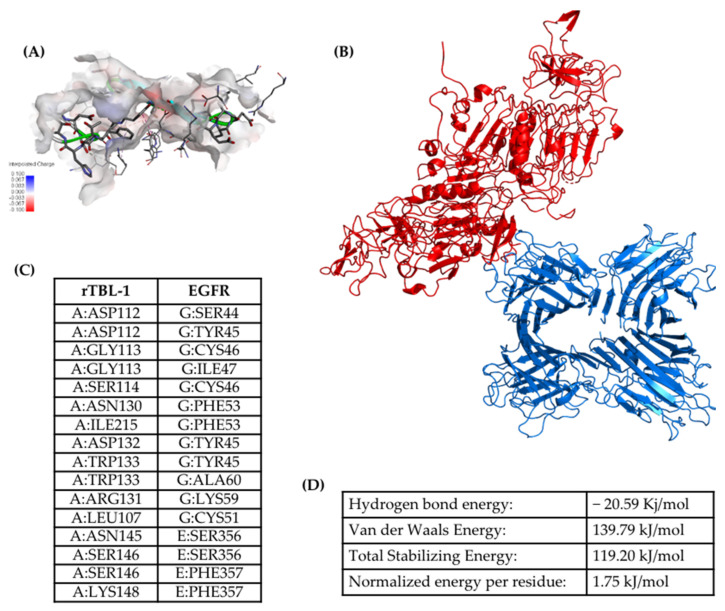
Docking of rTBL-1 and EGFR. (**A**) Topological representation of the protein–protein interaction zone showing EGFR-interacting residues as sticks. (**B**) The interaction between rTBL-1 and EGFR shows rTBL-1 and EGFR as blue and red ribbons, respectively. (**C**) Table with docking-interacting residues: A—Chain A of rTBL-1; E and G—Chain A and chain C of EGFR protein, respectively. (**D**) Standard energy calculation of the rTBL-1 and EGFR interaction measured by hydrogen bond energy and Van der Waals energy. Normalized energy per residue was obtained by dividing all non-bonded interactions by the total number of interface residues [44].

**Figure 9 pharmaceuticals-16-00290-f009:**
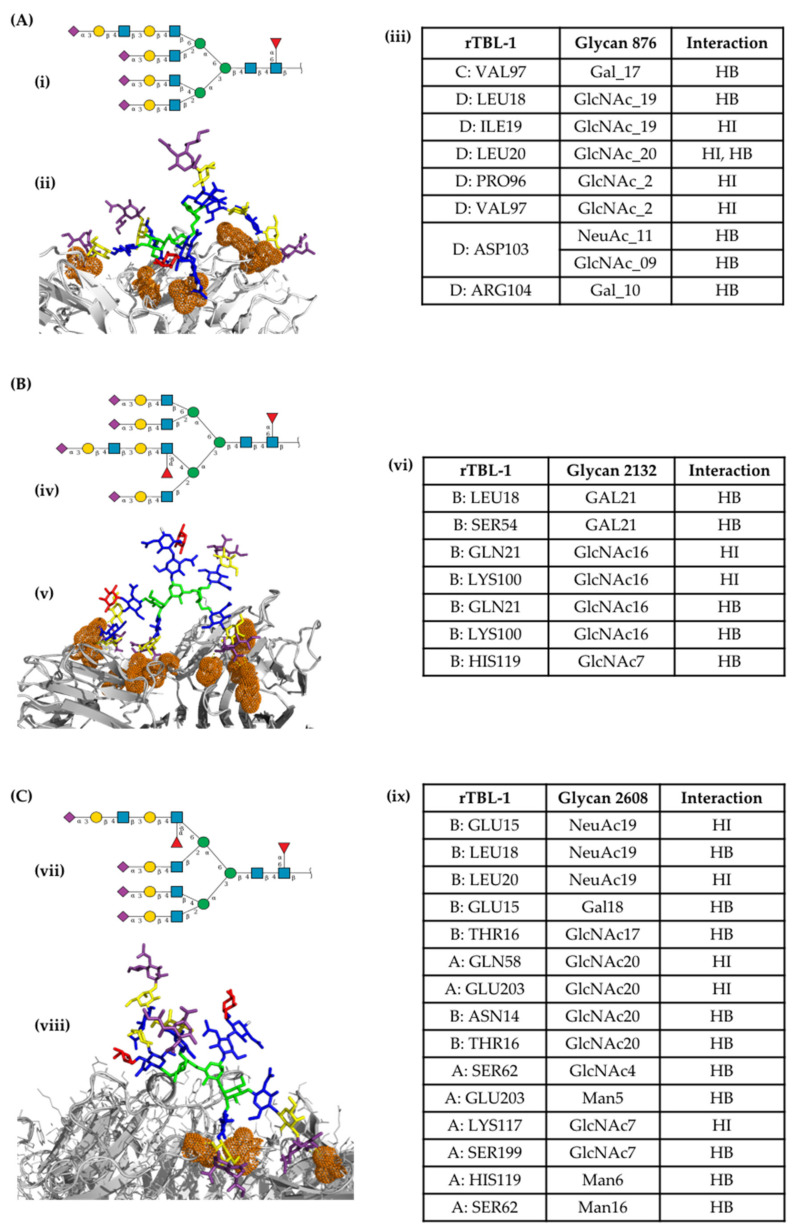
Molecular docking of recombinant Tepary bean lectin (rTBL-1) with EGFR-related glycans. (**A**) (i) Bidimensional representation of 876 N-glycan. (ii) Tridimensional representation of the interaction between 867-N-glycan and rTBL-1. (iii) Table with docking-interacting residues. (**B**) (iv) Bidimensional representation of 2132 N-glycan. (v) Tridimensional representation of the interaction between 2132-N-glycan and rTBL-1. (vi) Table with docking-interacting residues. (**C**) (vii) Bidimensional representation of 2608 N-glycan. (viii) Tridimensional representation of the interaction between 2608-N-glycan and rTBL-1. (ix) Table with docking-interacting residues. N-glycans are presented as sticks colored according to the symbol nomenclature for glycans (SNG) [46]. The rTBL-1 structure is shown as white ribbons, and interacting residues are highlighted with orange dots. The graphical 2D representation of the N-glycans structure is shown according to SNG. HI—Hydrophobic interaction. HB—Hydrogen bond. GlcNAc—N-Acetylglucosamine. Gal—Galactose. NeuAc—N-acetylneuraminic acid.

**Figure 10 pharmaceuticals-16-00290-f010:**
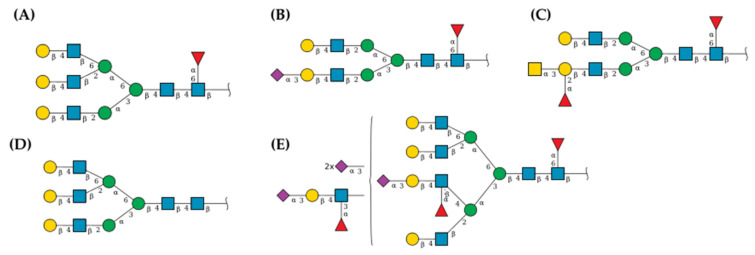
EGFR-related glycans with a high possibility of interaction with rTBL-1. N-glycans are presented according to the symbol nomenclature for glycans [46]. Docking data are not shown. (**A**) N-glycan 2263 with an average mass of 2152.9877. (**B**) N-glycan 2537 with an average mass of 2078.9082. (**C**) N-glycan 2874 with an average mass of 2136.9883. (**D**) N-glycan 3268 with an average mass of 2006.8447. (**E**) N-glycan 3414 with an average mass of 4340.9801.

**Figure 11 pharmaceuticals-16-00290-f011:**
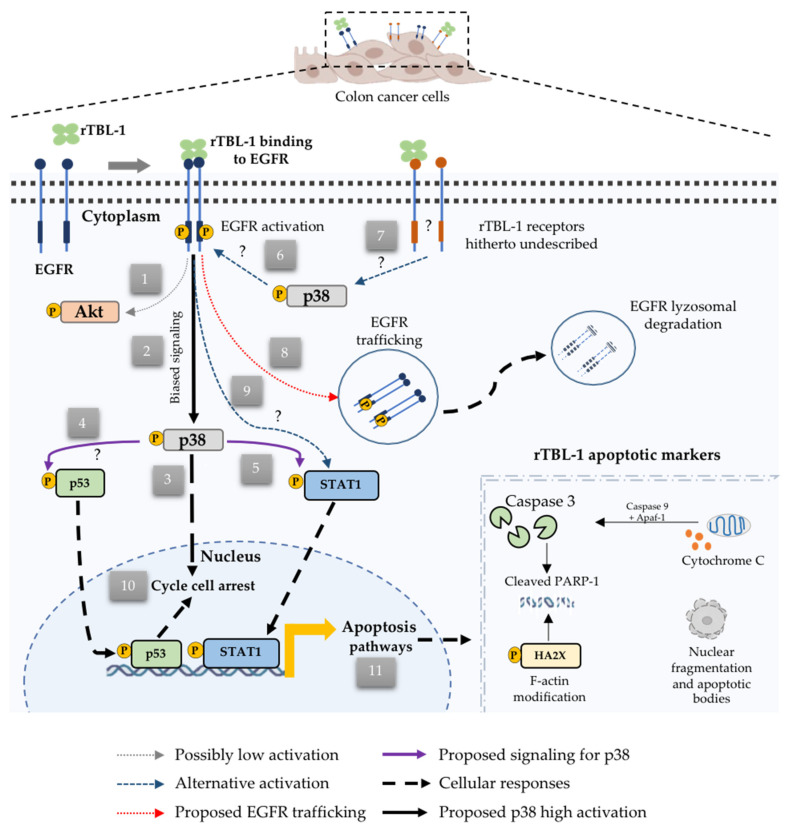
The proposed apoptotic mechanism for rTBL-1 through EGFR binding. rTBL-1 binds to EGFR or to hitherto undescribed receptors. (1) Low Akt activation inhibits cell proliferation and blocks inhibitory signaling for apoptosis. (2) Induction of EGFR-biased signaling provokes high activation of p38. (3) Translocation and signaling of p38. (4) Activation of p53. (5) Activation of STAT1. (6) Possible retroactive interaction between p38 and EGFR. (7) Possible activation of p38 by the interaction of rTBL-1 with other membrane receptors. (8) Partial lysosomal degradation of EGFR after the interaction with rTBL-1. (9) Other routes of STAT1 activation through rTBL-1-EGFR binding. (10) Effect on cell cycle arrest. (11) Identified apoptotic markers for rTBL-1 on human colon cancer cells. Figure was captured using the BioRender platform (https://biorender.com accessed on 28 December 2022.).

## Data Availability

Not applicable.

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
