# Peer review of "EGFR and p38MAPK Contribute to the Apoptotic Effect of the Recombinant Lectin from Tepary Bean (Phaseolus acutifolius) in Colon Cancer Cells"

_pharmaceuticals, 2023, doi:10.3390/ph16020290_

Round 1

Reviewer 1 Report

In this paper, you have presented data concerning the effect of rTBL-1 in colon cancer cells. A number of experiments performed to delineate the possible mechanism of action of rTBL-1. rTBL-1 binds to EGRF and subsequently increase the phosphorylation of p38, and other proteins leading to apoptosis of cancer cells. Experiments performed were well designed, described in details, well presented and analyzed, and thoroughly discussed.

Minor corrections:

Line 83: Other lectin to Other lectin,

Line 83: to internalize to to be internalized

Line 109: studies to studies,

Line 245: radio to ratio

Line 371: albalectin leaf lectins to alba leaf lectins

Author Response

Comments and Suggestions for Authors

In this paper, you have presented data concerning the effect of rTBL-1 in colon cancer cells. A number of experiments performed to delineate the possible mechanism of action of rTBL-1. rTBL-1 binds to EGRF and subsequently increase the phosphorylation of p38, and other proteins leading to apoptosis of cancer cells. Experiments performed were well designed, described in details, well presented and analyzed, and thoroughly discussed. 

Minor corrections:

Line 83: Other lectin to Other lectin

Correction was done in line 86.

Line 83: to internalize to to be internalized

Correction was done in line 86.

Line 109: studies to studies,

Correction was done in line 113.

Line 245: radio to ratio

Correction was done in line 231.

Line 371: albalectin leaf lectins to alba leaf lectins

Correction was done in line 422.

Thank you very much for your observations. The manuscript has already been reviewed by the English editing service.

Reviewer 2 Report

In this study, authors aimed to study the cytotoxic effect of rTBL-1 on colon cancer cells related to EGFR pathways. They showed rTBL-1 displayed similar apoptotic effects with TBLF for cleaving PARP-1 and caspase 3 and cell cycle G0/G1  arrest and decreasing S phase. They found rTBL-1 increased EGFR phosphorylation but also its degradation by the lysosomal route. Based on the comprehensive study, this work could be considered for publication in Pharmaceuticals after major revision.

-Two dots were used in Figure 1 (Line 132).

-Figure 3A is not clear.

-In line 280, “Apoptosis had also observed when othe….” Must be “Apoptosis had been also observed when othe….”.

-In Line 388, “fro” must be “for”.

-In Figure 8, if the figures were used or drawn with a program, it must be cited.

-Authors claim that lectin binds to the extracellular part of EGFR, however there is not any study confirming this hypothesis. Authors could perform in silico studies.

-There is not any toxicity report of lectins and recombinant lectin against healthy cells.

-The standard agents were not appropriate for EGFR studies.

-The advantage of use of recombinant form of lectin is not well-presented. Because the apoptotic parameters were found so close with lectin fraction.

-The references are appropriate.

Author Response

In this study, authors aimed to study the cytotoxic effect of rTBL-1 on colon cancer cells related to EGFR pathways. They showed rTBL-1 displayed similar apoptotic effects with TBLF for cleaving PARP-1 and caspase 3 and cell cycle G0/G1  arrest and decreasing S phase. They found rTBL-1 increased EGFR phosphorylation but also its degradation by the lysosomal route. Based on the comprehensive study, this work could be considered for publication in Pharmaceuticals after major revision.

-Two dots were used in Figure 1 (Line 132).

Correction was done.

-Figure 3A is not clear.

We changed Figure 3 in order to be clearer.

-In line 280, “Apoptosis had also observed when othe….”Must be “Apoptosis had been also observed when othe….”.

Correction was done in lines 331-332.

-In Line 388, “fro” must be “for”.

Correction was done

-In Figure 8, if the figures were used or drawn with a program, it must be cited.

Citation was included

-Authors claim that lectin binds to the extracellular part of EGFR, however there is not any study confirming this hypothesis. Authors could perform in silico studies.

We included an in silico analysis in lines 257-300, 435-447, 576-593, .

-There is not any toxicity report of lectins and recombinant lectin against healthy cells.

In previous works, it was considered to determine the effect of TBLF on non-cancer cells (García-Gasca et al., 2012), however, the availability of non-cancer human colon cell lines is limited. On the other hand, cell lines, even when they do not present a malignant phenotype, have a certain degree of transformation that allows them to adapt to in vitro conditions. The utility of in vitro systems is more related to the study of mechanisms of action, so the comparison between non-cancer cells against cancer cells for the identification of therapeutic molecules is of little use by itself if it is not related to specific molecular targets. Therefore, we have not included the comparison of the effect of rTBL-1 on non- colon cancer cells, but we didn’t detected apoptosis on CHO cells as in the present study shows, and this a non-cancer cell line. In previous studies we have compared the effect of TBLF on different human colon cancer cell lines according to their phenotypes (Moreno-Celis et al., 2020). In the present study, it was compared the apoptotic effect of rTBL-1 vs TBLF on HT-29 human colon cancer cells because they have shown to be more sensitive than other cell lines such as SW-480 and RKO. The study to determine the effect of rTBL-1 via EGFR was carried out in SW-480 cells given their high expression and CHO cells were taken as a negative control since they do not express the receptor.

-The standard agents were not appropriate for EGFR studies.

We are confused with this question, what do you mean with the standard agents? We did not use specific EGFR inhibitors in this work, in fact we are now working on it. The standard agents we had used were the specific inhibitors for proteasome and lysosomal pathways.

-The advantage of use of recombinant form of lectin is not well-presented. Because the apoptotic parameters were found so close with lectin fraction.

Indeed, one of the objectives of the work was to determine whether the apoptotic effect of rTBL-1 was similar to that of TBLF, as it turned out to be. The advantage of using rTBL-1 lies in the fact that TBLF is a semi-pure fraction of Tepary bean lectins, unlike rTBL-1, which is a single lectin. In addition, the process for obtaining TBLF is more expensive, low yields are obtained and takes longer. In contrast to rTBL-1, all these factors make the use of TBLF as a potential therapeutic agent not feasible, so the development of rTBL-1 with a therapeutic approach is necessary. We emphasized this argument in lines 603-607.

-The references are appropriate.

We revised all the references again.

Thank you very much for your observations. The manuscript has already been reviewed by the English editing service.

Round 2

Reviewer 2 Report

The authors have answered properly to all points and in my opinion the paper can now be published.